# Multiview Clickbait Detection via Jointly Modeling Subjective and Objective Preference

**Chongyang Shi[1], Yijun Yin[1], Qi Zhang[2†]**
**Liang Xiao[1], Usman Naseem[3], Shoujin Wang[4], Liang Hu[2]**
[1]Beijing Institute of Technology, China     [2]Tongji University, China
[3]James Cook University, Australia    [4]University of Technology Sydney, Australia
[1]{cy_shi, yin1jun, patrickxiao}@bit.edu.cn, [2]zhangqi_cs@tongji.edu.cn

## Abstract

Clickbait posts tend to spread inaccurate or misleading information to manipulate people's attention and emotions, which greatly harms the credibility of social media. Existing clickbait detection models rely on analyzing the objective semantics in posts or correlating posts with article content only. However, these models fail to identify and exploit the manipulation intention of clickbait from a user's subjective perspective, leading to limited capability to explore comprehensive clues of clickbait. Therefore, to bridge such a significant gap, we propose a multiview clickbait detection model, named MCDM, to model *subjective* and *objective* preferences simultaneously. MCDM introduces two novel complementary modules for modeling subjective feeling and objective content relevance, respectively. The subjective feeling module adopts a user-centric approach to capture subjective features of posts, such as language patterns and emotional inclinations. The objective module explores news elements from posts and models article content correlations to capture objective clues for clickbait detection. Extensive experimental results on two real-world datasets show that our proposed MCDM outperforms state-of-the-art approaches for clickbait detection, verifying the effectiveness of integrating subjective and objective preferences for detecting clickbait.

## 1 Introduction

Clickbait often creates inaccurate or misleading posts through the use of exaggeration or sensationalism to manipulate and exploit people's attention and emotions, rather than to inform or educate them (Molek-Kozakowska, 2013; Ng and Zhao, 2020). However, these posts often do not match their articles and can even be deceptive, misleading users' judgment of articles and harming the credibility of social media platforms. Accordingly, clickbait detection has become an emerging task for controlling the quality of social media content.

It aims at automatically detecting clickbait to help users quickly assess the credibility of the information on social media.

Recently, various clickbait detection models have been developed to identify clickbait articles, including traditional machine learning models and neural language models (Zheng et al., 2018; Glenski et al., 2017; Wu et al., 2020; Zhang et al., 2023). They analyze the posts and the corresponding article content to identify specific linguistic features and patterns, such as superlatives, exaggerations, emotive words, and incomplete sentences, typically associated with clickbait (Biyani et al., 2016; Wu et al., 2020). These models achieved desirable detection performance where users can avoid wasting time on misleading content, and social media platforms can maintain their credibility by providing accurate and informative content.

However, the complex and ever-changing tactics used by clickbait publishers bring great challenges to clickbait detection. Previous detection models generally neglect the manipulation intention of clickbait. In fact, it is necessary to obtain comprehensive clues about clickbait from both an *objective perspective* (i.e., the correlation between the corresponding content of articles and posts) and a *subjective perspective* (i.e., users' potential feelings about clickbait). As shown in Fig. 1, users impressed by the panic and shock in the post (caused by the terms "carry plastic" and "!") tend to focus on the post due to their subjective preference. When the users notify "Zimbabwean" and "U.S. dollar" in the article, they recognize the objective topic of the news, i.e., the devaluation of Zimbabwe's currency. Combing the two parts of information, users easily conclude that the post is out of context and exaggerated, showing that combining subjective and objective perspectives is necessary to capture comprehensive clues of clickbait and potentially improve detection performance.

In light of the above discussion, we propose a

---

† Corresponding author.

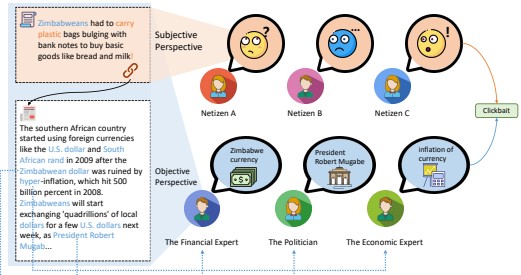

Figure 1: An illustration of subjective and objective information obtained from the post and its article. The red part and the blue part show users' subjective perspectives and objective perspectives, respectively.

multiview clickbait detection model (MCDM) via jointly modeling the subjective and objective preferences embodied in a post and its associated article. Specifically, the model contains two novel complementary modules for modeling subjective feeling and objective content relevance, respectively. The subjective module adopts BERT (Devlin et al., 2019) to obtain the semantics of a post, which captures subjective emotional and writing style features in a user-centric approach. The objective module introduces BERT and Longformer (Beltagy et al., 2020) to obtain post and article representations, respectively. It then adopts multi-head attention to fuse the post and article representations to capture objective clues. Our main contributions are summarized below:

- We propose a multiview clickbait detection model integrating subjective and objective preferences to capture comprehensive clues for clickbait detection.
- We model news element features in posts towards an objective perspective and apply a multi-head attention mechanism to extract contextual information in articles, which captures comprehensive clickbait clues in posts.
- Extensive experiments on two real-world datasets demonstrate the superior performance of our MCDM over state-of-the-art clickbait detection models, verifying the effectiveness of integrating subjective and objective preferences in detection.

## 2 Related Work

Clickbait detection has been constantly evolving and improving, from traditional methods based on manual features (Chakraborty et al., 2016; Rouvier and Favre, 2016) to automatic feature extraction using deep learning, as well as interaction and fusion of multiple features (Agrawal, 2016; Zhou, 2017; Zheng et al., 2018). The primary objective of clickbait detection is to extract crucial linguistic

features from textual content. Most of the current research is focused on modeling article content and detecting the correlation between posts and articles (Cao et al., 2017; Wei and Wan, 2017; Zheng et al., 2017), with the use of deep learning algorithms (Dong et al., 2018; Anand et al., 2017; Wu et al., 2020; Wei and Nguyen, 2022) increasingly employed to enhance the accuracy and efficiency of clickbait detection. Chen et al. (Chen et al., 2015) proposed the use of text and stylistic patterns for detection. Potthast et al. (Potthast et al., 2016) focused primarily on parts-of-speech features of posts, proposing a machine-learning model based on textual, structural, emotional, topical, and image features. Building upon previous work, Chakraborty et al. (Chakraborty et al., 2017) analyzed the social sharing patterns of clickbait and non-clickbait posts and utilized sentiment analysis tools to determine the emotional information possessed by each post. Zhou et al. (Zhou, 2017) employed a combination of bi-GRU network and self-attention mechanism to learn the post representation posted by users. However, relying solely on post information is insufficient, as longer articles can provide additional valuable information.

In recent years, more studies have focused on the correlation between posts and articles. Biyani et al. (Biyani et al., 2016) detected the correlation by using the similarity between the title and the first five sentences of the corresponding article. Zheng et al. (Zheng et al., 2017) proposed to recalculate the output of the traditional model according to user behavior. Wu et al. (Wu et al., 2020) considered the interaction between the context of posts and articles as well as the style patterns of posts. Wei et al. (Wei and Nguyen, 2022) proposed for the first time to use human semantic knowledge in an attention model and used language knowledge graphs to guide the attention mechanism. However, these methods only focus on considering the content correlation and neglect emotional and stylistic features that are crucial for clickbait detection. In contrast, our approach detects posts from subjective and objective perspectives jointly, facilitating accurate and quick clickbait detection. Furthermore, it can enhance representation by focusing on the objective correlation between posts and articles.

## 3 Problem Statement

The clickbait detection task is a binary text classification problem. Given a post on social media,

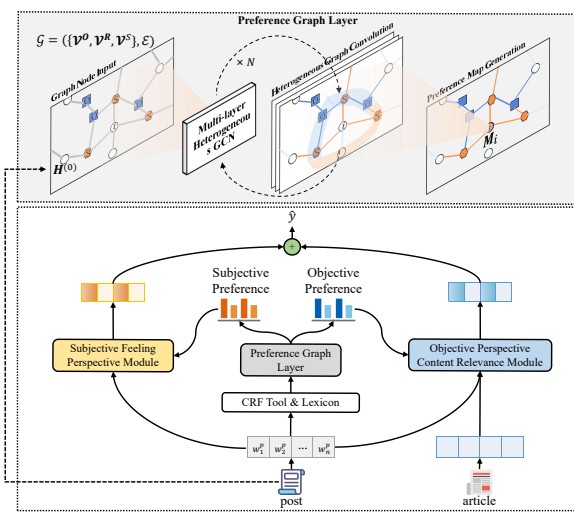

Figure 2: The architecture of our MCDM model.

represented as $\mathbf{P}_i = \{w_1, w_2, ..., w_n\}$ with $n$ representing the number of tokens, the corresponding article is represented as $\mathbf{D}_i = \{d_1, d_2, ..., d_m\}$ where $m$ is the number of tokens. From post $\mathbf{P}_i$, subjective feelings and objective entity knowledge are extracted to obtain a subjective preference map $\mathbf{M}^{sub} = [\mathbf{M}^{sub}_i]^n_{i=1}$ and an objective preference map $\mathbf{M}^{obj}$, with the $i^{th}$ fraction representing the model's degree of attention and preference for token $i$. The post text $\mathbf{P}$ is then used to extract its underlying emotional tendency feature $\mathbf{E}^P$, writing style feature $\mathbf{S}^P$, and initial semantic representation $\mathbf{H}^P$. Subsequently, the post, objective preference map, and corresponding article $\mathbf{D}_i$ are fed into the objective content association learning module to learn their interaction characteristics on the objective content, resulting in the objective content relation feature $\mathbf{G}^{obj}$. Finally, the shared feature module is used to obtain the clickbait representation $\boldsymbol{c} = [\mathbf{G}^{sub}, \mathbf{G}^{obj}]$, which is classified into a predefined category (i.e. clickbait/non-clickbait).

## 4 Framework

In this section, we introduce the framework of our proposed MCDM is shown in Fig. 2. It consists of four modules, i.e., a preference graph layer, a subjective feeling perspective modeling, an objective perspective content relevance modeling, and a perspectives fusion and output Layer.

### 4.1 Preference Graph Layer

This module extracts tokens that are biased toward subjective and objective perspectives respectively, generates preference scores for other tokens based on preference, and then dynamically adjusts the

maps of all tokens based on context. Following previous research (Sheng et al., 2021), subjective feeling lexicons are compiled from emotion and writing style knowledge of open source expert knowledge bases (Dong and Dong, 2003), matched with words in posts to obtain a collection of subjective feeling vocabulary. Similarly, the open source part-of-speech tagging tool TexSmart (Zhang et al., 2020; Liu et al., 2021) is used to extract entity information in posts and organize the vocabulary of proprietary nouns into a collection of objective entity vocabulary in the posts.

Specifically, in the undirected heterogeneous graph $\mathcal{G} = (\mathcal{V}, \mathcal{E})$ composed of tokens in post, there are three types of node sets: subjective feelings $\mathcal{V}^{\mathbf{S}}$, objective entities $\mathcal{V}^{\mathbf{O}}$, and other nodes (not biased towards these two perspectives) $\mathcal{V}^{\mathbf{R}}$. We extract the vocabulary node information of the graph, and use BERT to obtain the representations of the three types of vocabulary nodes. We concatenate them to initialize the heterogeneous graph $\mathbf{H}^{(0)} = [\mathbf{H}^{(0)}_{\mathcal{V}^{\mathbf{S}}}; \mathbf{H}^{(0)}_{\mathcal{V}^{\mathbf{O}}}; \mathbf{H}^{(0)}_{\mathcal{V}^{\mathbf{R}}}]$. Then, we initialize the weights of the connected edges between vocabulary nodes by cosine similarity, and the correlation matrix $A$ is normalized as

$$A^{(0)}(i,j) = \frac{\mathbf{H}^{(0)}_i \cdot \mathbf{H}^{(0)}_j}{2 \parallel \mathbf{H}^{(0)}_i \parallel_2 \parallel \mathbf{H}^{(0)}_j \parallel_2} + \frac{1}{2} \quad (1)$$

where $\mathbf{H}^{(0)}_i$ and $\mathbf{H}^{(0)}_j$ are initial node features, and $A^{(0)}(i,j) \in [0,1]$ is the initial weight of the edge connecting the $i$-th and $j$-th nodes. By multiplying the normalized degree matrix with the correlation matrix, we define the normalized correlation matrix $\hat{A}^{(l)}$ for layer $l$ as $\hat{A}^{(l)} = (D^{(l)})^{-\frac{1}{2}} A^{(l)} (D^{(l)})^{-\frac{1}{2}}$ where $D$ is the degree matrix.

To learn different perspectives of information, we use the heterogeneous dynamic GCN (Linmei et al., 2019; Ye et al., 2020). Specifically, we update the nodes of each layer with a dynamic correlation matrix and expect the weights of the final edges to reflect the preference of context nodes for different perspectives. For each layer of GCN, we use the correlation matrix $A$ and the state update weight matrix $W$ to update the value of the vocabulary node. Assuming the original input feature vector is $\mathbf{H}^{(0)}$, the $(l+1)$-th layer of GCN is

$$\mathbf{H}^{(l+1)} = \sigma(\sum_{\tau \in \mathcal{T}} \hat{A}^{(l)}_\tau H^{(l)}_\tau W^{(l)}_\tau) \quad (2)$$

where $\hat{A}^{(l)}_\tau$ is a submatrix of the $l$-th correlation matrix, whose rows contain all nodes, and columns

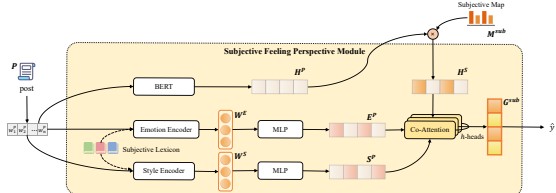

Figure 3: The architecture diagram of the SFP module.

record their correlation with nodes of type $\tau \in \{\mathcal{V}^{\mathbf{S}}, \mathcal{V}^{\mathbf{O}}, \mathcal{V}^{\mathbf{R}}\}$, $W_\tau \in \mathbb{R}^{d \times d'}$ is the weight matrix of type $\tau$ in this layer, $d'$ is the dimension of the output feature, $\sigma(\cdot)$ is the activation function, here using LeakyReLU to alleviate the problem of gradient disappearance in multi-layer networks. And the node correlation matrix $A$ is updated as

$$\triangle A^{(l+1)} = \sigma(\mathbf{H}^{(l+1)} W_A^{(l+1)} \mathbf{H}^{(l+1)T}) \quad (3)$$

$$A^{(l+1)} = \alpha A^{(l)} + (1-\alpha)\triangle A^{(l+1)} \quad (4)$$

where $W_A^{(l+1)}$ is the learnable weight matrix, $\sigma$ is the sigmoid function, and $\alpha$ is a weighting factor in [0,1]. We obtain the correlation matrix $A^{(L)}$ finally.

To obtain the subjective preference distribution $\mathbf{M}^{sub}$ for each token, we calculate the sum of the normalized correlation between the current node and all non-objective entity nodes. Similarly, the objective preference distribution $\mathbf{M}^{obj}$ is obtained following the same procedure.

## 4.2 Subjective Feeling Perspective Modeling

As shown in Fig. 3, this module excavates the subjectivity of posts from multiple perspectives such as writing style and emotional tendency.

Clickbait often contains emotionally or provocatively charged information. To comprehensively describe the emotional information contained in posts, this module extracts various emotional features (Zhang et al., 2021; Jiang et al., 2022), including emotion categories, emotion vocabulary, emotion intensity, emotion score, and other supplementary information, totaling 5 types of features, from post content. In addition to the information extracted from the emotional lexicon, a set of pattern information that captures non-lexical elements is introduced, including punctuation marks, emoticons, and writing habits description. For example, capital letters are used in a post to express stronger emotions, while emoticons such as ":)" can also replace vocabulary to express various emotions.

Meanwhile, special writing styles are also its typical means of expression, which distinguishes the good and bad quality of news and serves as an

important clue for detecting clickbait. According to the study of mining the quality of social news posts (Yang et al., 2019; Zhu et al., 2022), we extract 8 statistical features of writing styles, including readability, logic, credibility, formality, interactiveness, interestingness, user sensation, and integrity.

We extract subjective perception features from three aspects: semantic information, emotional tendency, and writing style. At the semantic level, we utilize the subjective preference map as attention weights to focus on vocabulary that is subjectively favored in the post. This approach generates a post-semantic representation $\mathbf{H}^S$ that combines with the subjective preference as

$$\mathbf{H}^S = \sum_{i=1}^{n} M_i^{sub} \mathbf{H}_i^P \quad (5)$$

By using emotional signals for modeling, five emotional signals extracted from the previous post are combined to obtain an overall emotional embedding vector $\mathbf{W}^E$, and then the emotional embedding vector $\mathbf{E}_i^P = \sigma(\mathbf{V}_E \mathbf{W}^E + b_E)$ is learned through a fully connected layer, obtaining the emotional feature representation $\mathbf{E}^P$, where $\mathbf{V}_E$ is a parameter matrix updated during network training, and $b_E$ is a bias term, $\sigma(\cdot)$ is the activation function ReLU.

Similarly, from the perspective of writing style, we first integrate the previously extracted writing style features in eight aspects into the style embedding vector $\mathbf{W}^S = \{s_1, ..., s_{|S|}\}$, and then use a fully connected layer with ReLU activation function to learn the style embedding vector $\mathbf{S}_i^P = ReLU(\mathbf{V}_S \mathbf{W}^S + b_S)$, obtaining the post style feature representation $\mathbf{S}^P$.

To comprehensively represent the subjective features of a post, the features from three perspectives of the post $\mathbf{P}$ are fused. Specifically, the vector obtained by concatenating the semantic, emotional, and stylistic features of the post $[\mathbf{H}^S, \mathbf{E}^P, \mathbf{S}^P]$ is linearly transformed and mapped to a Query vector matrix $\mathbf{Q}$, a Key vector matrix $\mathbf{K}$, and a Value vector matrix $\mathbf{V}$ respectively. Then, a multi-head attention mechanism is used to model and compute the weights of each element to be attended among multiple elements. Finally, the output vectors $\mathbf{z}^i$ of all attention heads are concatenated to obtain the multi-perspective features $\mathbf{G}^{sub}$ that comprehensively represent the subjective feeling.

## 4.3 Objective Content Relevance Modeling

This module focuses on the keywords related to the news elements, aiming to capture the correlation

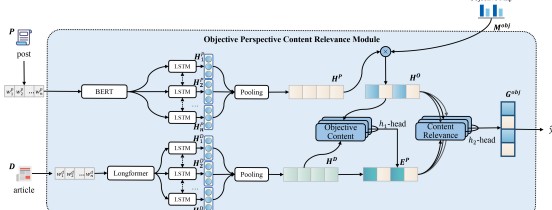

Figure 4: The architecture of the OPCR module.

between the posts and the corresponding article content from an objective perspective. The details of the module are shown in Fig. 4:

Through BiLSTM, the initial semantic representation $\mathbf{H}^P$ of the post is obtained, and the hidden state $\mathbf{h}_i^P \in \mathbb{R}^{2d}$ is calculated as $h_i^P = [\overleftarrow{\mathbf{h}_i}, \overrightarrow{\mathbf{h}_i}]$ where $\overrightarrow{\mathbf{h}_i}$ and $\overleftarrow{\mathbf{h}_i}$ are the hidden states in the forward and backward passes of BiLSTM, respectively. $d$ is the initial embedding dimension of the post.

To create an attention representation combining the objective entity features of the post, the weighted average is calculated based on the word-level scores of the post's objective entity map. The post representation $\mathcal{H}^P$ that focuses on objective content is obtained by the average pooling layer after the BiLSTM, which is expressed as

$$\mathcal{H}^P = \frac{1}{n} \sum_{i=1}^{n} \mathbf{h}_i^P \cdot \mathbf{M}_i^{obj} \quad (6)$$

We use Longformer to encode the article $\mathbf{D} = \{d_1, d_2, ..., d_m\}$ to obtain the initial semantic representation of the article, denoted as $\mathbf{H}^D$, and further encode it using BiLSTM to capture the content features of the post. The article is modeled as a matrix $\mathbf{H}'^D = [\mathbf{h}_1^D; \mathbf{h}_2^D; ...; \mathbf{h}_m^D] \in \mathbb{R}^{m \times 2d}$. We then guide the post to focus on the keywords in the article to establish a connection between the article and the post's objective entity information.

First, we replicate the post content representation vector $\mathcal{H}^P$ $m$ times and create a matrix $\mathcal{H}'^P \in \mathbb{R}^{m \times 2d}$. Second, the overall semantics of the article is usually obtained by its components collectively. Therefore, we propose an attention mechanism for focusing on important words in the article by extending the vector $w_2$ to the matrix $W_2 \in \mathbb{R}^{a_1 \times h_1}$ as follows:

$$A_1 = \text{Softmax}(\tanh([\mathbf{H}'^D; \mathcal{H}'^P] \cdot W_1) \cdot W_2) \quad (7)$$

where $A_1 \in \mathbb{R}^{m \times h_1}$, $h_1$ is the number of attention heads, and each column of $A_1$ is normalized by Softmax. After computing $A_1$, the objective content representation of the article $\mathcal{H}^D$ is obtained:

$$\mathcal{H}^D = \text{Flatten}(A_1^T \cdot \mathbf{H}'^D) \quad (8)$$

where $\mathcal{H}^D \in \mathbb{R}^{h_1 \times 2d}$, and the function Flatten($\cdot$) flattens $A_1^T \cdot \mathbf{H}'^D$ into a vector.

We use a collaborative attention network to capture the content-related relationship between posts and articles, where different attention distributions represent different degree of clickbait of articles for posts, thereby ensuring their deep interaction:

$$\mathbf{H} = \text{Attention}(\mathbf{Q}, \mathbf{K}, \mathbf{V}) \quad (9)$$

where the matrix $\mathbf{Q} = \mathcal{H}^D$, $\mathbf{K} = \mathbf{V} = \mathcal{H}^P$, and $d_k = 2d$ is the size of the BiLSTM hidden units.

To enhance network ability, the multi-head attention uses different linear projections of the query $\mathbf{Q}$, key $\mathbf{K}$ and value $\mathbf{V}$, projecting them $h_2$ times, and then performing parallel scaled dot-product attention. Finally, the processed results are concatenated and projected to obtain a new representation:

$$\mathbf{G}^{obj} = \text{Multihead}(\mathbf{Q}, \mathbf{K}, \mathbf{V}) \quad (10)$$

where the global attention of content-relatedness applies $h_2$ different heads, and $\mathbf{W}_i^Q \in \mathbb{R}^{2d \times d_1}$, $\mathbf{W}_i^K \in \mathbb{R}^{2d \times d_1}$, $\mathbf{W}_i^V \in \mathbb{R}^{2d \times d_1}$, $\mathbf{W}^O \in \mathbb{R}^{2d \times 2d}$ are trainable parameters, with $d_1 = \frac{2d}{h_2}$. Finally, we obtain the content-related interaction features $\mathbf{G}^{obj}$ that contain objective entity information.

## 4.4 Perspectives Fusion and Output Layer

We combine the representation of subjective multi-perspective posts and objective content relevance to obtain the final clickbait representation $\boldsymbol{c} = [\mathbf{G}^{sub}; \mathbf{G}^{obj}]$, and the classification category $\hat{y} = \text{MLP}(\boldsymbol{c})$ is calculated. Then, we train our model by minimizing the cross-entropy loss $\mathcal{L}_{cls}(y, \hat{y})$.

Besides, we use the cosine similarity between subjective sentiment and objective entity preference maps, to guide the classification to be more closely aligned with specific subjective and objective perspectives, an auxiliary loss is introduced:

$$\mathcal{L}_{sim}(\mathbf{M}^{sub}, \mathbf{M}^{obj}) = \frac{\mathbf{M}^{sub} \cdot \mathbf{M}^{obj}}{\parallel \mathbf{M}^{sub} \parallel_2 \parallel \mathbf{M}^{obj} \parallel_2} \quad (11)$$

In addition, we flip the actual labels and exchange the subjective and objective preference maps input, introducing a signal of reverse supervision to emphasize the current preference result. Accordingly, another auxiliary loss is employed:

$$\mathcal{L}_{cls}(y_{con}, \hat{y}') = \text{CELoss}(y_{con}, \hat{y}') \quad (12)$$

where the flipped label $y_{con} = \mid 1 - y \mid$, and the predicted result $y' = \text{MLP}(\boldsymbol{c}')$ where $\boldsymbol{c}'$ is the preference maps with the opposite viewpoints used as

output. We obtain the final objective function:

$$\mathcal{L} = \mathcal{L}_{cls} + \alpha_1 \mathcal{L}_{sim} + \alpha_2 \mathcal{L}_{cls}(y_{con}, \hat{y}') \quad (13)$$

where $\alpha_1$ and $\alpha_2$ are weighting factors that measure the degree of involvement of map similarity and flipped labels. Since the overall loss function is dominated by the final classification loss of the model, their values range is in [0.1, 0.5].

## 5 Experiments and Analysis

### 5.1 Experimental Setup

**Datasets:** In our experiment, we utilized two real-world datasets, namely Clickbait Challenge17[1] and FNC Challenge[2]. Both datasets contain posts and articles. The Clickbait Challenge17 dataset comprises 33,000 samples collected from 27 major news agencies in the United States. Similar to (Dong et al., 2018), we labeled each pair of posts and article samples with a clickbait tag if their average score was above 0.5. The FNC Challenge dataset consists of 89,000 samples. Following previous studies (Dong et al., 2018; Wu et al., 2020), we classify "unrelated" items as clickbait.

**Methods:** We compared MCDM to nine SOTA baselines and divided the baselines into two groups: *Using only posts:* **DSSM** (Huang et al., 2013): A deep network is used to obtain hidden features of the input and quantify the similarity in the potential representation space. **CLSM** (Shen et al., 2014): A variant of DSSM, uses convolutional neural networks to extract intrinsic semantic features. **CBCNN** (Zheng et al., 2018): A clickbait detection model based on textCNN (Chen, 2015).
*Using both posts and articles:* **LiNN** (Glenski et al., 2017): A model using LSTM and CNN networks to learn the vectorized text and visual content, respectively. **MSA** (Kumar et al., 2018): The attention-based bidirectional RNN method is used to learn the input data, and then the latent information and relational knowledge are combined into the Siamese network. **BiGRU-Att** (Zhou, 2017): Attention-based BiGRU model that focuses on latent features contained in the semantics of a post and its linked articles. **LSDA** (Dong et al., 2018): The model uses GRU-Att to measure global and local similarities between post and article representation vectors. **SATC** (Wu et al., 2020): A detection method that is sensitive to post style and considers synergistic attention between posts and articles.

---

[1] https://webis.de/events/clickbait-challenge/
[2] http://www.fakenewschallenge.org/

**KED** (Wei and Nguyen, 2022): A method of integrating human semantic knowledge into a neural network model, by building a language knowledge graph to guide attention mechanisms for inference.

**Implementation Details:** In the experiments, we used Bert to initialize the word embedding layer, each with a dimension of 768 dimensions and Longformer to initialize the article embedding. The GCN network was initialized with glorot, while the classification layer of BERT was initialized with Xavier normal. The learning rate was set to 0.0001, and the size of each mini-batch is 8. The maximum sequence length for posts is 100, and 600 for articles. We conducted 20 iterations and set the output unit dimension of BiLSTM to 128. In the SFP module, we utilized shared learning of the multi-head attention mechanism with 3 heads. To avoid overfitting, we set the dropout rate to 0.1. For the OPC module, we set the number of attention heads for learning the content of articles to 5 and for global content-related attention to 2. Additionally, we used AdamW as our optimizer, and the weight decay value for L2 regularization was set to 0.0001. These key parameters were tuned on the validation set. We use four metrics, i.e., accuracy, precision, recall, and F1-score, for evaluation.

### 5.2 Performance Evaluation

Table 1 presents the overall performance of all the comparison methods. According to the results, we have several main findings.

1) The methods that incorporate article and post information (BiGRU-Att, SATC, KED) outperform those that solely rely on post information (DSSM, CLSM, CBCNN), emphasizing the importance of jointly detecting post and article information.

2) The methods that utilize attention to capture the interaction between articles and posts (e.g., BiGRU-Att, LSDA) perform better than those without attention (e.g., CBCNN, LiNN). This is because the attention mechanism highlighting important contexts in both posts and articles allows for better modeling of their association in clickbait detection.

3) The methods that consider clickbait emotional and stylistic patterns (SATC, KED) show superior performance over most other methods, demonstrating their strong ability to learn post representations and capture subjective sentiments.

4) Our MCDM performs better compared to the baselines. We attribute its superiority to three advantages: (1) through the update of the preference

Table 1: Performance of MCDM and baselines on datasets.

| Methods | Clickbait Challenge17 | | | | FNC Challenge | | | |
|---|---|---|---|---|---|---|---|---|
| | Accuracy | Precision | Recall | F1 | Accuracy | Precision | Recall | F1 |
| DSSM (Huang et al., 2013) | 0.817 | 0.655 | 0.661 | 0.658 | 0.747 | 0.894 | 0.740 | 0.811 |
| CLSM (Shen et al., 2014) | 0.833 | 0.683 | 0.643 | 0.662 | 0.756 | **0.959** | 0.762 | 0.853 |
| CBCNN (Zheng et al., 2018) | 0.844 | 0.654 | 0.653 | 0.653 | 0.789 | 0.852 | 0.845 | 0.857 |
| LiNN (Glenski et al., 2017) | 0.827 | 0.642 | 0.621 | 0.631 | 0.868 | 0.925 | 0.884 | 0.913 |
| MSA (Kumar et al., 2018) | 0.826 | 0.699 | 0.474 | 0.565 | 0.859 | 0.920 | 0.884 | 0.913 |
| BiGRU-Att (Zhou, 2017) | 0.856 | 0.719 | 0.650 | 0.683 | 0.879 | 0.924 | 0.897 | 0.919 |
| LSDA (Dong et al., 2018) | 0.860 | 0.722 | 0.699 | 0.710 | 0.894 | 0.933 | 0.912 | 0.918 |
| SATC (Wu et al., 2020) | 0.869 | 0.762 | 0.712 | 0.737 | 0.907 | 0.929 | 0.903 | 0.916 |
| KED (Wei and Nguyen, 2022) | 0.873 | 0.761 | 0.745 | 0.753 | 0.918 | 0.927 | 0.913 | 0.925 |
| SFP | 0.865 | **0.787** | 0.745 | 0.766 | 0.916 | 0.936 | 0.918 | 0.927 |
| OPCR | 0.873 | 0.776 | 0.749 | 0.764 | 0.922 | 0.941 | 0.922 | 0.936 |
| MCDM | **0.881** | 0.782 | **0.754** | **0.768** | **0.927** | 0.958 | **0.922** | **0.945** |

Table 2: Performance comparison of MCDM variants.

| MCDM variants | Clickbait17 | | FNC | |
|---|---|---|---|---|
| | Acc. | F1 | Acc. | F1 |
| SFP - post | 0.836 | 0.751 | 0.891 | 0.896 |
| SFP - submap | 0.854 | 0.759 | 0.909 | 0.919 |
| SFP - emotion | 0.858 | 0.755 | 0.882 | 0.893 |
| SFP - style | 0.853 | 0.752 | 0.901 | 0.918 |
| SFP | 0.865 | 0.766 | 0.916 | 0.927 |
| OPCR - post | 0.842 | 0.740 | 0.903 | 0.927 |
| OPCR - objpost | 0.853 | 0.755 | 0.897 | 0.913 |
| OPCR - p & a | 0.862 | 0.768 | 0.916 | 0.929 |
| OPCR | 0.873 | 0.764 | 0.922 | 0.936 |

graph network for posts, it can obtain more comprehensive subjective and objective feature representations of post content; (2) the introduction of subjective sentiment features allows for capturing potentially clickbait textual cues from a subjective perspective; (3) combining news entity preference features improves the model's ability to handle scenarios with unclear subjective sentiments and focuses on the interaction between posts and articles, allowing for better modeling of their joint representations from an objective perspective.

## 5.3 Ablation Studies

In this section, we conduct ablation studies on the SFP model by investigating the usage of initial post semantics, subjective preference features, emotional features, and writing style features, as well as on the OPCR model by exploring post-article encoding, objective entity preference learning, and content correlation learning modules, to demonstrate the effectiveness of each module's design.

The experimental results, as shown in Table 2, reveal that in the subjective perception module ablation experiment, SFP performs better than several variants in both datasets, indicating the effectiveness of the subjective perception features selected in this study. The SFP-post variant has the worst performance, suggesting that not delving into other features of posts cannot enhance the experimental results. The SFP-submap has the best performance among several variants, confirming the importance of capturing semantic and writing style features in comprehensively characterizing subjective preference, and learning subjective preference in posts optimizes the detection of clickbait.

In the ablation experiment, the overall performance of the OPCR model is higher than that of the other variants, indicating the effectiveness of the feature combinations selected in this study. In both datasets, the experimental effects of OPCR-post variant and OPCR-objpost variant are not as good as that of the OPCR-post & article variant, which means that the effect of only using post features is inferior to that of using both post and article features, demonstrating the effectiveness of using post and article information simultaneously, and validating that the use of more relevant text features improves the detection of clickbait. Therefore, each module proposed in the SFP and OPCR models plays a critical role in improving the overall performance of the models.

## 5.4 Qualitative Analysis

We analyze the impact of post length on performance. The experimental results, shown in Fig. 5, indicate a significant improvement when the post length increases from 30 to 100 words Although the F1-score is slightly higher for the 150-word post, the overall improvement is not substantial. The threshold parameter $n$ is set to 100 for the Clickbait Challenge17 dataset and 150 for the FNC

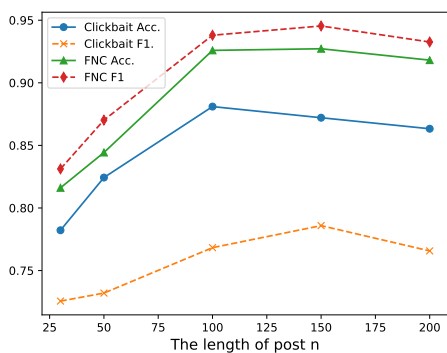

Figure 5: Performance of the MCDM model under different values of the parameter $n$.

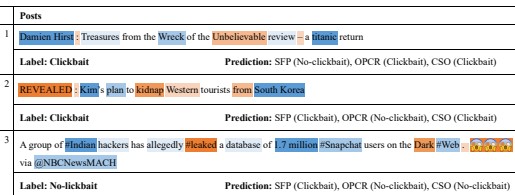

| | Posts |
|---|---|
| 1 | Damien Hirst : Treasures from the Wreck of the Unbelievable review – a titanic return |
| | **Label: Clickbait**      **Prediction:** SFP (No-clickbait), OPCR (Clickbait), CSO (Clickbait) |
| 2 | REVEALED : Kim's plan to kidnap Western tourists from South Korea |
| | **Label: Clickbait**      **Prediction:** SFP (Clickbait), OPCR (No-clickbait), CSO (Clickbait) |
| 3 | A group of #Indian hackers has allegedly #leaked a database of 1.7 million #Snapchat users on the Dark #Web 😱😱😱 via @NBCNewsMACH |
| | **Label: No-lickbait**      **Prediction:** SFP (Clickbait), OPCR (No-clickbait), CSO (No-clickbait) |

Figure 6: Three examples of clickbait and non-clickbait posts. The red markers indicate the vocabulary of subjective feelings, while the blue markers indicate the vocabulary of objective entities. The darker the color, the higher the preference score for that particular word.

Challenge dataset, considering the character limit of 140 words in Twitter for the former dataset. Taking into account time and memory constraints, we choose 100 as the post length.

We further investigate the effectiveness of $\alpha_1$ and $\alpha_2$ in balancing the contribution of preference map loss and inversion label loss. Table 3 shows the experimental results. Increasing $\alpha_1$ (larger than 0.3) harms MCDM's effectiveness, while increasing $\alpha_2$ significantly enhances MCDM's overall performance. The optimal performance is achieved at $\alpha_1 = 0.3$ and $\alpha_2 = 0.5$. As $\alpha_2$ increases, the weight of the inversion label loss in the total loss rises, allowing MCDM to focus more on distinguishing preferences opposite to its own. This distinction between subjective and objective perspectives improves overall performance. By appropriately weighing different perspectives in the loss function, controlled by $\alpha_1$, MCDM can better adjust to different preference maps, enhancing its performance and generalization ability.

Table 3: Detection accuracy of MCDM on Clickbait Challenge17 dataset with different $\alpha_1$ and $\alpha_2$.

| | | $\alpha_1$ | | | | |
|---|---|---|---|---|---|---|
| | | 0.1 | 0.2 | 0.3 | 0.4 | 0.5 |
| | 0.1 | 0.722 | 0.858 | 0.859 | 0.864 | 0.871 |
| | 0.2 | 0.841 | 0.869 | 0.867 | 0.880 | 0.876 |
| $\alpha_2$ | 0.3 | 0.874 | 0.875 | 0.877 | 0.879 | **0.881** |
| | 0.4 | 0.864 | 0.866 | 0.871 | 0.824 | 0.877 |
| | 0.5 | 0.862 | 0.860 | 0.868 | 0.873 | 0.876 |

## 6 Case Studies

We evaluate the model's classification accuracy by extracting subjective and objective preference features. The SFP model and the OPCR model are employed to map attention scores for subjective and objective perspectives, respectively, on click-bait and non-clickbait posts. The MCDM model's overall success rate in classification, combining both subjective and objective perspectives, is analyzed. Fig. 6 presents three posts successfully classified by the MCDM, SFP, and OPCR models, demonstrating the avoidance of biases commonly observed in clickbait detection through the fusion of subjective and objective features.

In the first case, the post's sentiment was weak, leading the SFP model to misclassify it as non-clickbait, while the OPCR model correctly identified its lack of actual relationship with the article and categorized it as clickbait. In the second case, the post exhibited strong emotional bias and writing style signals, accurately recognized by the subjective preference SFP model. However, the OPCR model mistakenly classified it as non-clickbait due to the presence of content appearing in the article. In the third post, the OPCR model effectively detected and correctly classified the extensive objective entity information. However, the SFP model erroneously classified it as clickbait due to the presence of emoticons and emotionally biased adjectives. These cases highlight MCDM's efficacy in incorporating both subjective and objective feature associations for clickbait detection.

## 7 Conclusion

In this paper, we propose MCDM, a clickbait detection framework guided by subjective and objective perspectives. MCDM takes into account the emotional and stylistic characteristics of posts from the subjective viewpoint and the clickbait information from the objective perspective by combining both post and article content. Extensive experiments on two real-world datasets demonstrate that MCDM effectively improves the detection performance of clickbait by combining the features of posts from the subjective level and the interaction between posts and articles on the objective entity level.

## Limitations

Although extensive experiments have demonstrated that MCDM can effectively improve the performance of the model, as mentioned above, there are limitations to consider. Firstly, both perspectives' prior information relies on manually designed features, whereas an automated detection process based on text content, independent of lexicons and tools, is desired. Additionally, our current model only focuses on clickbait detection based on textual information, neglecting the rich subjective and objective information conveyed through images accompanying the posts. Thus, our model lacks analysis of image content. This will also be a direction of improvement for our future work.

## Acknowledgements

This work is supported by National Natural Science Foundation of China (No. 62102026).

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
