# OpenReview forum: "Multiview Clickbait Detection via Jointly Modeling Subjective and Objective Preference"
_EMNLP/2023/Conference — EMNLP 2023 Findings_

### Official Review · Reviewer_7Cfx · 2023-08-01

**Soundness:** 4

**Excitement:**

4: Strong: This paper deepens the understanding of some phenomenon or lowers the barriers to an existing research direction.

**Missing References:**

None

**Paper Topic And Main Contributions:**

The authors propose a clickbait detection model called MCDM to address the significant gap in clickbait detection. MCDM consists of two novel modules: a subjective feeling module and an objective content relevance module. The subjective feeling module captures user-centric features of posts, including language patterns and emotional inclinations. On the other hand, the objective module examines news elements and models article content correlations to identify objective clues for clickbait detection. The experimental results on two real-world datasets demonstrate that MCDM outperforms state-of-the-art approaches for clickbait detection. This confirms the effectiveness of integrating subjective and objective preferences in detecting clickbait content.

**Questions For The Authors:**

Can you include more recent baseline methods?

**Reasons To Accept:**

1. The idea of modelling subjective feeling for clickbait detection is novel and clever.

2. The proposed method to utilize both subjective feeling and objective content relevance is solid and reasonable.

3. The experiments are comprehensive and convincing.

**Reasons To Reject:**

1. It would be better if more real-world datasets are used in experiments.

2. There are a few flaws in paper writing. First, the font in many figures is too small, making it difficult to read. Second, there should be some punctuation at the end of the equations.

**Reproducibility:**

4: Could mostly reproduce the results, but there may be some variation because of sample variance or minor variations in their interpretation of the protocol or method.

**Reviewer Confidence:**

4: Quite sure. I tried to check the important points carefully. It's unlikely, though conceivable, that I missed something that should affect my ratings.

**Typos Grammar Style And Presentation Improvements:**

First, the font in many figures is too small, making it difficult to read. Second, there should be some punctuation at the end of the equations.

---

> ### Author Rebuttal · Authors · 2023-08-29
>
> We much appreciate your time and valuable comments. We provide the following explanations to address your concerns.
>
> **1. It would be better if more real-world datasets are used in experiments.**
>
> Thank you for your suggestion. Due to time limitations, it is hard for us to report the results of all comparative methods on new real-world datasets. But, we will perform experiments on new real-world datasets, e.g., News Clickbait Detection (News) [4] and News Headline Incongruence Detection (NELA) [4], and report the corresponding results in the final version.
>
> In this work, our selection of datasets in this work is inspired by [1, 2, 3]. Both the datasets we utilized are sourced from real-world Twitter and corresponding news websites, collected by data providers without any filtering or processing.
>
> [1] An Attention-Based Neural Network Using Human Semantic Knowledge and Its Application to Clickbait Detection
>
> [2] Clickbait detection with style-aware title modeling and co-attention, CCL, 2020.
>
> [3] Similarity-aware deep attentive model for clickbait detection, PAKDD, 2019.
>
> [4] Clickbait Detection via Contrastive Variational Modelling of Text and Label, IJCAI 2022.
>
> **2. There are a few flaws in paper writing. First, the font in many figures is too small, making it difficult to read. Second, there should be some punctuation at the end of the equations.**
>
> Thank you for your valuable suggestions.
>
> + Firstly, we apologize for the inconvenience caused by the small font size in certain figures, which may have affected the readability. We will redraw the figures and ensure that the text and labels are clearly legible, enhancing the overall quality of the visuals.
>
> + Secondly, we understand the importance of proper punctuation at the end of equations. In the revised version of the paper, we will ensure that all equations are appropriately punctuated for clarity and accuracy.
>
> We assure you that we will thoroughly review and polish the paper to guarantee its overall quality. We are dedicated to making these improvements and delivering an enhanced final version.
>
> **3. Can you include more recent baseline methods?**
>
> Yes, we will include three recent baseline methods, i.e., LSACD [7], RCD [5], and CVM [6], in the experiments. We report their performance on the two datasets in the following table.
>
> | | |Clickbait |Challenge17  | | | FNC | Challenge | |
> |:---|:---|:---|:---|:---|:---|:---|:---|:---|
> | Methods| Accuracy| Precision| Recall| F1|Accuracy| Precision| Recall| F1|
> |LSACD [7]| 0.846 | 0.742 | 0.698 | 0.797 | 0.913 | 0.836 | 0.847 | 0.825|
> |RCD [5]| 0.857 | 0.769 | 0.756 | 0.791 | 0.862 | 0.772 | 0.824 | 0.743|
> |CVM [6]| 0.860 | - | - | 0.694 | - | - | - | -|
> |MCDM(ours) | **0.881*** | **0.782*** | **0.754*** | **0.768*** | **0.927*** | **0.958*** | **0.922*** | **0.945*** |
>
> From the table, we can observe that our proposed MCDM outperforms the recent baselines, indicating the effectiveness of our proposed model.
>
> Thank you for your suggestion. We will report the corresponding results in the final version to deliver a more convincing experiment evaluation.
>
> [5] Rumor and clickbait detection by combining information divergence measures and deep learning techniques, ARES 2022.
>
> [6] Clickbait Detection via Contrastive Variational Modelling of Text and Label, IJCAI 2022.
>
> [7] A deep model based on Lure and Similarity for Adaptive Clickbait Detection, KBS 2021.
>
>
> Thank you for your positive comments and valuable suggestions. We hope our explanations can address all your concerns.

---

### Official Review · Reviewer_FUSD · 2023-08-06

**Typos Grammar Style And Presentation Improvements:** NA
**Soundness:** 3

**Excitement:**

4: Strong: This paper deepens the understanding of some phenomenon or lowers the barriers to an existing research direction.

**Missing References:**

NA

**Paper Topic And Main Contributions:**

This paper is about clickbait detection on social media, which is an important task in practice. It presents a multi-view clickbait detection model, which can capture both subjective and objective preference features to address the problem. The proposed method can employ the emotional characteristics of posts from the subjective viewpoint and the clickbait features from the objective perspective via anlayzing both post and article content. The work conducts extensive experiments on the two real-life datasets, and shows that the proposed method outperforms the selected baselines. This paper was well written and is easy to follow.


**Questions For The Authors:**

1) This work aims to address clickbait detection task, and formulates it as a text classification problem. In practice it would be more interesting to explore multimodal data sources such as images for the problem.
2) In Section 5.1, it would be good to describe more about the process of annotating clickbait tags for instances.
3) In Table 1, it's not clear if the results of proposed method are statistically significant compared to baselines such as KED.


**Reasons To Accept:**

1) This paper presents a multiview clickbait detection model, which can employ both subjective and objective preference features for the clickbait prediction.
2) The work relies on a multi-head attention strategy to extract contextual information in articles, which can capture the comprehensive clickbait clues in posts.


**Reasons To Reject:**

In practice, it may lead to limited generalizability to rely on manually defined list of features and preferences, or selected lexicons, especially applying the proposed method across social media platforms.

**Reproducibility:**

4: Could mostly reproduce the results, but there may be some variation because of sample variance or minor variations in their interpretation of the protocol or method.

**Reviewer Confidence:**

3: Pretty sure, but there's a chance I missed something. Although I have a good feel for this area in general, I did not carefully check the paper's details, e.g., the math, experimental design, or novelty.

---

> ### Author Rebuttal · Authors · 2023-08-29
>
> We much appreciate your time and valuable comments. We provide the following explanations to address your concerns.
>
> **Regarding generalizability**
>
> We appreciate your thoughtful consideration of the potential limitation that our proposed model relies on manually designed features, which we have also discussed in **Section Limitation**. We recognize that this limitation may pose challenges and difficulties in terms of the applicability and adaptability of our model when deployed across social media platforms.
>
> Furthermore, we would like to emphasize that our model has been meticulously crafted to address the need for adaptability to manual features across different platforms.
>
> + Flexible Model Architecture: Our model architecture is flexibly designed to accommodate different manually defined features and preferences as well as incorporate domain-specific lexicons for improved performance on different platforms, ensuring our model has the potential to be effectively applied across social media platforms.
>
> + Adaptive Learning: Our model incorporates several advanced learning modules, such as dynamic Graph Convolutional Networks (GCN) and Multi-Head Attention. These modules empower our model to identify and adapt to various linguistic nuances and emerging trends prevalent in diverse social media platforms.
>
> In conclusion, we sincerely appreciate the thoroughness with which you have reviewed and considered our proposed model. We will fully consider the potential limitation regarding the model's applicability across social media platforms and assure you that we are committed to addressing this limitation in our future work. We recognize the significance of studying cross-platform or cross-language click-bait detection models as a captivating avenue for future research.
>
> **Q1: In practice it would be more interesting to explore multimodal data sources such as images for the problem.**
>
> We highly value your valuable suggestion, which sheds light on an interesting direction. It is true that incorporating multimodal data sources, including images, has the potential to enhance clickbait detection, a point we have also addressed in **Section Limitations** of our discussion. In fact, considering the multimodal nature of clickbait is a topic that we plan to focus on and improve in our future endeavors.
>
> In addition, the existing benchmark datasets for clickbait detection currently suffer from a shortage of image data. For instance, the FNC Challenge dataset does not include any images, and although the Clickbait Challenge17 datasets do contain some images, they represent only a small fraction, approximately one quarter, of the total samples. The lack of sufficient image data poses significant challenges for the clickbait detection community in effectively incorporating multimodal data sources. Most recent methods for clickbait detection primarily emphasize the analysis of textual information. Accordingly, we did not consider multimodal clickbait detection models in this work, and we are dedicated to expanding our future research in this direction to advance the field of clickbait detection.
>
>
> **Q2: In Section 5.1, it would be good to describe more about the process of annotating clickbait tags for instances.**
>
> Thanks for your valuable suggestion. Considering the page limitations, we have made a deliberate decision to prioritize providing detailed explanations of our model to provide a better understanding and ensure complete reproducibility of our model for future researchers. Note that the two datasets in our experiments are widely used and publicly available for clickbait detection, both of which come with pre-annotated labels.
>
> Herein, we provide a thorough study and analysis of these datasets:
> + FNC Challenge dataset: The dataset comprises three parts: headline, body, and labels. Building upon the existing labels {unrelated, discuss, agree, disagree}, we followed [1,2] to further classify them into {clickbait, non-clickbait}. Specifically, we defined "unrelated" as clickbait and the remaining three stances as non-clickbait in accordance with the dataset's description.
> + Clickbait Challenge 17 dataset: As described in its paper [3] and related blogs, to mitigate publisher and topical biases, a maximum of ten posts per day and per publisher were sampled. All posts were annotated on a 4-point scale [not click baiting (0.0), slightly click baiting (0.33), considerably click baiting (0.66), heavily click baiting (1.0)] by five annotators from Amazon Mechanical Turk.
>
> We will supplement these details in the final version. We believe that by including this information in Section 5.1, we enhance the transparency of our dataset annotation process and facilitate a better understanding of the two datasets.
>
> [1] An Attention-Based Neural Network Using Human Semantic Knowledge and Its Application to Clickbait Detection. IEEE Open J. Comput. Soc. 3: 217-232 (2022).
>
> [2] Clickbait detection with style-aware title modeling and co-attention. ACL 2020.
>
> [3] Crowdsourcing a Large Corpus of Clickbait on Twitter. COLING 2018.
>
> **Q3: In Table 1, it's not clear if the results of the proposed method are statistically significant compared to baselines such as KED.**
>
> In Table 1, we compared the baselines with our proposed MCDM and its two variants (SFP and OPCR):
> + The Subjective Feelings Preference-based Clickbait Detection Model (SFP) leverages subjective sentiment cues such as emotional tendencies and writing style to identify posts with inducing (clickbait) characteristics.
> + The Objective Content Relationship-based Clickbait Detection Model (OPCR) utilizes entity information related to the post's news elements and combines it with corresponding article content.
> + The Model Combining Subjective and Objective Perspectives (MCDM) integrates both subjective and objective viewpoints.
>
> From Table 1, we can observe that MCDM outperforms almost all baselines (including the state-of-the-art baseline KED) on both two datasets in terms of all four metrics (accuracy, precision, recall, and F1 score). For example, we show the corresponding results of our MCDM and KED in the following table (copied from Table 1). Herein, we highlight the **statistically significant improvement** (i.e., **two-sided t-test with p < 0.05**) indicated by $*$. These impressive results show significant advantages of our proposed MCDM over KED. Also, MCDM achieved more substantial improvement over other clickbait detection models. Our MCDM's advantages stem from its incorporation of three main features: comprehensive subjective and objective content representations, subjective sentiment features, and news entity preference feature that enhance the model's ability to handle scenarios with unclear subjective sentiments.
>
> | | |Clickbait |Challenge17  | | | FNC | Challenge | |
> |:---|:---|:---|:---|:---|:---|:---|:---|:---|
> | Methods| Accuracy| Precision| Recall| F1|Accuracy| Precision| Recall| F1|
> |KED | 0.873 | 0.761 | 0.745 | 0.753 | 0.918 | 0.927 | 0.913 | 0.925|
> |MCDM(ours) | **0.881*** | **0.782*** | **0.754*** | **0.768*** | **0.927*** | **0.958*** | **0.922*** | **0.945*** |
>
> In summary, our model's performance improvement over the baselines is multi-faceted and comprehensive, indicating the significant superiority of MCDM over the baselines.
>
> Thank you for your positive comments and valuable suggestions. We hope our explanations can address all your concerns.

---

### Official Review · Reviewer_mx4P · 2023-08-09

**Typos Grammar Style And Presentation Improvements:** None
**Soundness:** 3

**Excitement:**

4: Strong: This paper deepens the understanding of some phenomenon or lowers the barriers to an existing research direction.

**Missing References:**

None

**Paper Topic And Main Contributions:**

This paper proposes a multi-view titled party detection model (MCDM) to address the problem that existing models fail to identify and mine titled party manipulation intentions from the perspective of user agents. MCDM introduces two new complementary modules for modeling subjective perception and objective content relevance, respectively. The subjective sentiment module adopts a user-centered approach to capture the subjective characteristics of posts, such as language patterns and emotional inclusion. The targeting module explores news elements from posts and models article content relevance to capture targeting cues for headline party detection. Extensive experimental results on two real datasets demonstrate that MCDM outperforms the state-of-the-art methods in headline fraud detection.

**Questions For The Authors:**

A . The second contribution presented in the paper confuses the reader, as the relevance of titles to articles has become a widespread concern in many past studies. Therefore, it may be difficult for readers to understand exactly what is the innovation of this paper in this respect.

**Reasons To Accept:**

1. MCDM introduces two new complementary modules to model subjective perception and objective content association, respectively. Compared with the baseline method, MCDM comprehensively considers the cues about headline party and improves the detection performance.

**Reasons To Reject:**

1. In the introduction, the description of the motivation and purpose of the research is not comprehensive enough, and there is a lack of a comprehensive elaboration of the framework of the article.
2. The baseline method compared in the article is somewhat outdated, and there is only one method after 21 years. This situation leads to a lack of sufficient convincing arguments for the validity of the model.
3. Compared with the baseline method, the model proposed in the paper does not show significant improvement in overall accuracy, especially on the Clickbait Challenge17 dataset. Although the author may have made some innovations and improvements in the model design and experiment process, from the experimental results, these improvements did not significantly improve the performance of the model.
4. Many outdated references were cited, with only four from 2021 onwards. This also proves that the method proposed in the article is not advanced.

**Reproducibility:**

3: Could reproduce the results with some difficulty. The settings of parameters are underspecified or subjectively determined; the training/evaluation data are not widely available.

**Reviewer Confidence:**

4: Quite sure. I tried to check the important points carefully. It's unlikely, though conceivable, that I missed something that should affect my ratings.

---

> ### Author Rebuttal · Authors · 2023-08-29
>
> We sincerely appreciate you taking the time and providing valuable comments. In the following, we provide a detailed response to address all of your concerns.
>
> **1. Regarding the motivation and purpose of the research and the framework of the article**
>
> + It is essential to recognize that clickbait often utilizes inaccurate or misleading tactics to manipulate and exploit people's attention and emotions, thus distorting their perception of information and undermining the credibility of social media platforms. As a result, clickbait detection has emerged as a crucial and practical task for maintaining the quality of social media content.
>
>     In our research, we are motivated by the realization that achieving a comprehensive clickbait detection requires an understanding of both the emotional and stylistic elements (subjective) as well as the relevance of the objective content (objective). Our objective is to develop a more reliable approach for automatically detecting clickbait by integrating both subjective and objective perspectives. By doing so, we aim to enhance the effectiveness of clickbait detection and mitigate its negative impact on users and social media platforms.
>
> +  Regarding the framework of our article, we will enhance the paragraph preceding the contribution summarization in the introduction and provide a more comprehensive introduction of the framework. Specifically, we will
>     - highlight how the subjective and objective perspectives are integrated to enhance the accuracy of clickbait detection,
>     - supplement more details of each module in our framework to show how we construct the network architecture in the framework.
>
> We believe that these revisions will address the shortcomings you mentioned and provide a more comprehensive and structured introduction.
>
>
> **2. Regarding baseline method**
>
> Thank you for your valuable comments. In this work, we initially chose representative baselines of each category for the clickbait detection evaluation to guarantee a comprehensive comparison between our proposed MCDM with existing clickbait detection models. In addition, we compared our MCDM with the state-of-the-art baseline (KED [6] published at the end of 2022 which is the current state-of-the-art in the field and served as our latest baseline) to show the state-of-the-art performance of MCDM and verify the superiority of our proposed model.
>
> Furthermore, we have analyzed some recent literature, i.e., LSACD [14], RCD [10], and CVM [12], and compared our MCDM with the three baselines during rebuttal. We report their performance on the two datasets in the following table where ∗ indicates the statistically significant improvement (i.e., two-sided t-test with p < 0.05). We will complete the experiments and include the corresponding results in the final version.
>
> | | |Clickbait |Challenge17  | | | FNC | Challenge | |
> |:---|:---|:---|:---|:---|:---|:---|:---|:---|
> | Methods| Accuracy| Precision| Recall| F1|Accuracy| Precision| Recall| F1|
> |LSACD [14]| 0.846 | 0.742 | 0.698 | 0.797 | 0.913 | 0.836 | 0.847 | 0.825|
> |RCD [10]| 0.857 | 0.769 | 0.756 | 0.791 | 0.862 | 0.772 | 0.824 | 0.743|
> |CVM [12]| 0.860 | - | - | 0.694 | - | - | - | -|
> |MCDM(ours) | **0.881*** | **0.782*** | **0.754*** | **0.768*** | **0.927*** | **0.958*** | **0.922*** | **0.945*** |
>
> To address your concerns, we will discuss baselines [10,12,14] in the related work and include the baselines in the experiments.
>
> **3. Regarding performance improvement**
>
> Firstly, we want to emphasize that our datasets are collected from real-world scenarios, which often result in uneven data distribution. As such, in the context of this classification task, our focus extends beyond merely improving accuracy; it includes placing substantial importance on the enhancement of the F1 score. This metric is particularly significant for addressing imbalanced data and capturing the trade-off between precision and recall, which is a common characteristic of clickbait detection.
>
> In Table 1, we compared the baselines with our proposed MCDM and its two variants (SFP and OPCR):
> + The Subjective Feelings Preference-based Clickbait Detection Model (SFP) leverages subjective sentiment cues such as emotional tendencies and writing style to identify posts with inducing (clickbait) characteristics.
> + The Objective Content Relationship-based Clickbait Detection Model (OPCR) utilizes entity information related to the post's news elements and combines it with corresponding article content.
> + The Model Combining Subjective and Objective Perspectives (MCDM) integrates both subjective and objective viewpoints.
>
> From Table 1, we can observe that MCDM outperforms almost all baselines (including the state-of-the-art baseline KED) on both two datasets in terms of all four metrics (accuracy, precision, recall, and F1 score). For example, we show the corresponding results of our MCDM and KED in the following table (copied from Table 1). Herein, we highlight the **statistically significant improvement** (i.e., **two-sided t-test with p < 0.05**) indicated by $*$. These impressive results show significant advantages of our proposed MCDM over KED. Also, MCDM achieved more substantial improvement over other clickbait detection models. Our MCDM's advantages stem from its incorporation of three main features: comprehensive subjective and objective content representations, subjective sentiment features, and news entity preference feature that enhance the model's ability to handle scenarios with unclear subjective sentiments.
>
> | | |Clickbait |Challenge17  | | | FNC | Challenge | |
> |:---|:---|:---|:---|:---|:---|:---|:---|:---|
> | Methods| Accuracy| Precision| Recall| F1|Accuracy| Precision| Recall| F1|
> |KED | 0.873 | 0.761 | 0.745 | 0.753 | 0.918 | 0.927 | 0.913 | 0.925|
> |MCDM(ours) | **0.881*** | **0.782*** | **0.754*** | **0.768*** | **0.927*** | **0.958*** | **0.922*** | **0.945*** |
>
> In summary, our model's performance improvement over the baselines is multi-faceted and comprehensive, indicating the significant superiority of MCDM over the baselines.
>
>
> **4. Regarding outdated references**
>
> To address your concerns about the outdated references, we would like to provide a comprehensive analysis of recent literature, and we will add more recent references in the final version.
>
> + Since our paper was submitted on June 16, 2023, there were only three articles related to clickbait detection listed in 2023:
>     - Reference [1] focuses on the study involving a Turkish language dataset.
>     - Reference [2] primarily contributes to addressing clickbait editing decisions within the competitive context of platform regulation instead of clickbait click models.
>     - Reference [3] explores detecting clickbait by utilizing ChatGPT's response to questions
>     The three models are unsuitable for experimental comparison.
>
> + In the landscape of 2022 publications related to clickbait detection, only nine relevant works were formally published. Some of these studies can be summarized as follows, and others even omitted critical dataset and task details, making it challenging to perform a comprehensive comparison:
>     - Reference [4] approaches clickbait from the perspective of blockchain research, focusing primarily on evaluating detection performance from security and memory angles.
>     - Reference [5] employs a Chinese dataset and integrates semantic, syntactic, and auxiliary information for clickbait detection.
>
> + Note that there have not been quite a few studies on clickbait detection recently. The reason for the relatively limited number of recent references could be as follows:
>     - This research task is relatively niche, being a subfield of fake news detection, but it holds significant real-world importance in purifying the atmosphere of social networks. Much of the research in this area was conducted in 2017 and 2018 after the release of the Clickbait Challenge 17 competition and dataset.
>     - In recent years, the NLP field has attracted more attention to Natural Language Generation (NLG), and there might be relatively fewer studies on classification tasks and even clickbait detection.
>
> We have taken your suggestion seriously and conducted in-depth analyses of valuable recent literature [7,8,10,12,14]. We will incorporate references to works published after 2021 [1,2,3,7,8,10,12,14] in the Related Work section to make our study more comprehensive and persuasive.
>
>
> **Regarding the second contribution**
>
> Thank you for your comments. Due to space constraints, the articulation of our contributions might have led to some confusion. We apologize for any ambiguity that may have arisen from our previous description. To address this, we have revised and reorganized our wording to provide you with a clearer and more comprehensive explanation of the second contribution:
> + Our method stems from the foundation of examining objective entity elements within posts. By harnessing the inherent attributes of news entity elements, which posts tend to embrace from an objective perspective, and by synergizing this with contextual information from the associated articles, we aim to more precisely capture the implicit triggering cues embedded within posts. This approach is meticulously designed to circumvent misinterpretations stemming from subjective biases.
> + This innovation aligns seamlessly with our motivation to explore the dual facets of subjective and objective perspectives. Specifically, we leverage extracted entity elements as auxiliary information for posts, incorporating a dynamic Graph Convolutional Network (GCN) to capture and underscore the objective dimensions of post features related to news entities. By harnessing the entity information correlated to the news elements in posts and incorporating corresponding article content, our model adopts a human-like in-depth analysis of the objective perspective of clickbait content. We extract features associated with the objective entity content relevance and create comprehensive representations of both the post and its corresponding article. This approach stands in stark contrast to previous studies, distinguishing our work and methodology from existing literature.
>
> We hope that our elaboration clarifies the uniqueness and technical significance of our second contribution. We will polish the statement of the second contribution accordingly in the final version.
>
> [1] Artificial intelligence on the advance to enhance educational assessment: Scientific clickbait or genuine gamechanger?. Journal of Computer Assisted Learning, 2023.
>
> [2] A game theory framework of the relation between journalism, users, and platforms. New Media & Society, 2023.
>
> [3] Clickbait Detection via Large Language Models. arXiv 2023.
>
> [4] Blockchain-enabled deep recurrent neural network model for clickbait detection. IEEE Access 2021.
>
> [5] Clickbait detection on WeChat: A deep model integrating semantic and syntactic information. KBS, 2022.
>
> [6] An Attention-Based Neural Network Using Human Semantic Knowledge and Its Application to Clickbait Detection. IEEE Open Journal of the Computer Society, 2022.
>
> [7] Detecting Clickbait in Chinese Social Media by Prompt Learning. CSCWD 2023.
>
> [8] Clickbait spoiler type identification with transformers. Proceedings of the 17th International Workshop on Semantic Evaluation (SemEval-2023). 2023.
>
> [10] Rumor and clickbait detection by combining information divergence measures and deep learning techniques. Proceedings of the 17th International Conference on Availability, Reliability and Security. 2022: 1-6.
>
> [11] Clickbait detection in tweets using self-attentive network. arXiv 2017.
>
> [12] Clickbait Detection via Contrastive Variational Modelling of Text and Label. IJCAI 2022.
>
> [13] Clickbait Headline Detection in Indonesian News Sites using Multilingual Bidirectional Encoder Representations from Transformers (M-BERT). arXiv 2021.
>
> [14] A deep model based on lure and similarity for adaptive clickbait detection. KBS 2021.
>
> [15] Clickbait detection with style-aware title modeling and co-attention. CCL 2020.
>
>
> Thank you for your valuable feedback, which has driven us to refine our presentation and enhance the clarity of our work.

---

### Meta-Review · Area_Chair_owG3 · 2023-09-17

**Recommendation:** 3

**Metareview:**

The reviewers saw both strengths and weaknesses in the submitted version of the paper. The authors have provided a rebuttal that appeared to alleviate some concerns. After some discussion, the overall verdict was leaning toward an acceptance of this submission. However, there are still some issues to be addressed as raised by Reviewer mx4P. So, a suggestion was made to accept the paper as Findings. The reviewers' concerns should be considered in the final version.

---

### Decision · Program_Chairs · 2023-10-07

**Decision:**

Accept-Findings

**Comment:**

The reviewers saw both strengths and weaknesses in the submitted version of the paper. The authors have provided a rebuttal that appeared to alleviate some concerns. After some discussion, the overall verdict was leaning toward an acceptance of this submission. However, there are still some issues to be addressed as raised by Reviewer mx4P. So, a suggestion was made to accept the paper as Findings. The reviewers' concerns should be considered in the final version.